# Immunochemical characterisation of styrene maleic acid lipid particles prepared from *Mycobacterium tuberculosis* plasma membrane

Sudhir Sinha[1]*, Shashikant Kumar[1], Komal Singh[1], Fareha Umam[1], Vinita Agrawal[2], Amita Aggarwal[1], Barbara Imperiali[3]

1 Department of Clinical Immunology & Rheumatology, Sanjay Gandhi Postgraduate Institute of Medical Sciences, Lucknow, India, 2 Department of Pathology, Sanjay Gandhi Postgraduate Institute of Medical Sciences, Lucknow, India, 3 Department of Biology, Massachusetts Institute of Technology, Cambridge, Massachusetts, United States of America

* sinha.sudhir@gmail.com

**Data Availability Statement:** All relevant data are within the paper and its Supporting information files.

## Abstract

Membrane proteins of *Mycobacterium tuberculosis* (Mtb) can be targeted for the development of therapeutic and prophylactic interventions against tuberculosis. We have utilized the unique membrane-solubilising properties of the styrene maleic acid copolymer <styrene:maleic acid::2:1> (SMA) to prepare and characterise 'styrene maleic acid lipid particles' from the native membrane of Mtb (MtM-SMALPs). When resolved by SDS-PAGE and visualised with coomassie blue, the molecular weights of Mtb membrane (MtM) proteins solubilised by SMA were mostly in the range of 40–70 kDa. When visualised by transmission electron microscopy, MtM-SMALPs appeared as nanoparticles of discrete shapes and sizes. The discoid nanoparticles exhibited a range of diameters of ~10–90 nm, with largest portion (~61%) ranging from 20–40 nm. MtM proteins of a molecular weight-range overlapping with that of MtM-SMALPs were also amenable to chemical cross-linking, revealing protein complex formation. Characterisation using monoclonal antibodies against seven MtM-associated antigens confirmed the incorporation of the inner membrane protein PRA, membrane-associated proteins PstS1, LqpH and Ag85, and the lipoglycan LAM into MtM-SMALPs. Conversely, the peripheral membrane proteins Acr and PspA were nearly completely excluded. Furthermore, although MtM showed an abundance of Con A-binding glycoproteins, MtM-SMALPs appeared devoid of these species. Immune responses of healthcare workers harbouring 'latent TB infection' provided additional insights. While MtM-SMALPs and MtM induced comparable levels of the cytokine IFN-γ, only MtM-SMALPs could induce the production of TNF-α. Antibodies present in the donor sera showed significantly higher binding to MtM than to MtM-SMALPs. These results have implications for the development of MtM-based immunoprophylaxis against tuberculosis.

**Funding:** The authors received no specific funding for this work.

**Competing interests:** The authors have declared that no competing interests exist.

## Introduction

Tuberculosis (TB) is caused by an airborne infection with *Mycobacterium tuberculosis* (Mtb) [1]. An estimated 10.6 million people developed TB and 1.6 million died from it in 2021 [2]. In the backdrop of the COVID-19 pandemic, the number of people with undiagnosed and untreated TB has increased, leading to an escalation in TB deaths. In addition, the burden of drug-resistant TB has also increased [2]. Treatment of TB involves dosing with several potentially toxic drugs for protracted periods. Even then the treatment success rate is not very impressive, being nearly 85% for the drug-sensitive and 57% for drug-resistant TB. BCG (Bacille Calmette-Guerin), the only licensed vaccine against TB is generally ineffective in adults although it confers some degree of protection in children [2]. In this situation, the need for new and more efficient means of diagnosis, treatment or prevention of TB cannot be overstated.

Membrane proteins carrying one or more α-helical transmembrane domains constitute 20–30% of the proteome from all genomes and perform a wide range of biological functions [3, 4]. Over 60% of all drug targets are membrane proteins and some of the frontline TB drugs target the biogenesis of Mtb membrane (MtM) [5, 6]. In addition, the mycobacterial membrane proteins can elicit potent immune responses in persons exposed to Mtb [7–9]. Membrane proteins are difficult to express or purify in a stable and biologically active form. Therefore, despite their abundance and biological importance, relatively few membrane proteins have been purified and characterized to date. Just 2.3% of all depositions in the protein data bank (PDB) are of membrane proteins, even though there has been an exponential growth in membrane protein structures solved during the past five years [10, 11]. The proteome of Mtb comprises 3993 predicted proteins [12, 13], of which 682 are predicted to be membrane proteins with at least one trans-membrane helix. However, so far only about 7% of these proteins have entries in the PDB.

The application of amphipathic copolymers of styrene and maleic acid (SMA) represents a major advance in the study of membrane proteins. SMA copolymers solubilise biological membranes by forming SMA lipid particles (SMALPs) which remove membrane proteins along with a significant quantity of surrounding lipids, thereby ensuring the retention of native membrane protein folds [14–19]. SMALPs are the only system capable of extracting membrane proteins without an intermediate detergent solubilisation step, while preserving their native environment. This unique feature ensures better stability of proteins within the SMALPs and also makes SMA highly unlikely to disturb the protein-protein interactions that result in the formation of large membrane protein complexes [20]. It can therefore be assumed that the SMALPs conserve the structure as well as function of the membrane proteins. The proteins in SMALPs essentially behave like soluble proteins, amenable to biochemical and biophysical characterisation. Another unique advantage of SMALPs is that both faces of an embedded protein (or protein complex) are accessible to the corresponding ligands or substrates. SMALP-based methods have been successfully employed for the analysis of ion channels, transporters, enzymes, respiratory complexes and receptors; all of which were extracted from a wide range of biological membranes [15, 21].

According to the studies performed with model membranes [15–17], the first step in SMALP formation is binding of SMA copolymer to the surface of membrane lipid bilayer. In the next step, the aryl groups (derived from styrene monomers) of the SMA polymer bury deeper into the hydrophobic core of the membrane. In this process, the aryl moieties intercalate between the lipid acyl chains. The charged maleic acid groups are oriented in the same direction as the aryl moieties but interact with the lipid headgroups. Essentially, the SMA copolymer assumes the shape of a 'bracelet' encircling a patch of the membrane lipid bilayer.

In the final step, membrane fragments are further solubilised to generate SMALP nanoparticles. In this sequence of events culminating in supramolecular assembly of SMALPs, a crucial step is the burial of aryl groups into hydrophobic core of the membrane. As SMA copolymers do not discriminate between different lipids, the native lipid environment of membrane proteins encased within SMALPs is retained. Nonetheless, lipid bilayer properties such as fluidity, thickness, lateral pressure and charge density play distinct roles in the kinetics of solubilisation. Lipid packing strongly influences the efficiency of solubilisation and unsaturated lipids have been found to be more difficult to solubilise than the saturated ones. Due to the double bonds in the acyl tails, the unsaturated lipids exert an enhanced lateral pressure in the acyl-chain region of the lipid bilayer which may lead to a less efficient insertion of SMA.

We have previously reported the purification and characterisation of membrane proteins from Mtb and other mycobacteria using biphasic separation with Triton X-114, preparative SDS-PAGE and proteomics [7, 22–24]. However, those attempts had met with a limited success with respect to the identification and characterisation of trans-membrane helix-bearing membrane proteins as a majority of the identified proteins proved to be 'peripheral membrane proteins' [25]. In this study, considering the unique membrane-solubilising ability of SMA copolymers, we have prepared and immunochemically characterised the SMALPs from the whole native membrane of Mtb. In addition, we have performed functional characterisation of the MtM-SMALPs in terms of their ability to induce cell-mediated and humoral (antibody) immune responses in healthcare workers harbouring a 'latent TB infection (LTBI)' [26]. We have also performed a subset of experiments with the lyophilised MtM-SMALPs. SMALPs are currently being developed for clinical applications [27, 28] and a lyophilised preparation (if it shows an activity comparable with the 'wet' SMALPs) can be stored and transported with considerable ease, without a cold chain.

## Materials and methods

### Study subjects

Six healthy healthcare workers, who were positive for the 'tuberculin skin test' [26, 29] signifying the presence of LTBI, were enrolled for the study. None of the study subjects had clinical or radiological evidence suggestive of TB or any other disease. Their blood samples were collected by standard venipuncture in sodium-heparin tubes (1 mL, for T cell and macrophage function assays) and in plain tubes (3 mL, to get sera for antibody assays). The study protocol was approved by Institutional Ethics Committee of Sanjay Gandhi Postgraduate Institute of Medical Sciences (SGPGIMS), Lucknow (IEC approval code: 2016-149-IMP-EXP). All experiments were performed in accordance with Declaration of Helsinki Guidelines (https://www.wma.net) and all blood donors provided a written informed consent.

### Mtb cell membrane

MtM was isolated by using a previously described method [7]. In brief, culture of Mtb (strain H37Ra, ATCC25177) in Lowenstein-Jensen medium was harvested and bacteria were washed and re-suspended in tris buffer (50 mM Tris-HCl, pH 8.0). The cell lysate obtained by probe-sonication was centrifuged, initially at 23,000g to settle unbroken cells and cell debris, and later at 150,000g to obtain the cell membrane as sediment and cytosol as supernatant. Membrane was washed and reconstituted in the tris buffer. All protein estimations were done by a modified Lowry's method suitable for membrane proteins [8] and aliquots were stored at -80°C. The taxonomic identity of Mtb was verified by immune-chromatographic detection of MPT64 antigen [8] using a kit ('SD Bioline TB Ag MPT64 Rapid', Abbott, USA).

## MtM-SMALPs

SMA (styrene: maleic acid:: 2: 1) was prepared according to a previously described protocol [14]. To prepare MtM-SMALPs, SMA stock (10% w/v in tris buffer, pH 8.0) was added to MtM (10 mg protein in tris buffer) to get the final SMA concentration of 2.5%. The tube was incubated (2 h at room temperature, RT) with occasional tapping. Later, the suspension was centrifuged (92,500g, 1 h, 24˚C) using Beckman Type 25 fixed angle rotor. The supernatant (containing SMALPs) was collected and the sediment (insoluble membrane proteins) was reconstituted with tris buffer. Protein was estimated in both the fractions and aliquots were stored at -80˚C. For comparison, the following variations of the protocol were also tried [14]: (a) Overnight (at RT) extraction of MtM with SMA in tris buffer and (b) extraction (2 h, RT) of MtM with SMA in tris buffer containing 150 mM NaCl.

To detect any free (unbound to SMA) protein, the MtM-SMALP suspension (0.5 mL) was filtered through a 100 kDa cutoff filter (Spin-X UF 500, Corning, USA) following the manufacturer's protocol. The filtrate was collected after centrifugation and the residual top layer was washed three times, each time by reconstituting in 0.5 mL tris buffer and centrifuging as per protocol. Protein was estimated in the first filtrate and final top layer. The choice for 100 kDa cutoff was based on the observed radius of the smallest MtM-SMALPs (~5 nm, as determined by electron microscopy, see below) and the empirical evidence that the Stoke's radius of 5 nm corresponds to a molecular mass of ~200 kDa [30].

To determine the effect of lyophilization on their structure and immunochemical properties, an aliquot of MtM-SMALPs was lyophilized and reconstituted with water prior to use.

In addition to SMA, Triton X-100 (TX-100) was also used for the extraction of membrane proteins [31]. TX-100 (2% final concentration) was added to MtM and incubated overnight (at RT) with end-to-end mixing. After centrifugation (92,500g, 1 h), TX-100 soluble supernatant was collected and the insoluble sediment was reconstituted with tris buffer. To determine its solubility in SMA, the TX-100 insoluble fraction was re-extracted (2 h, RT) with SMA as described above. SMALPs were collected and the SMA-insoluble sediment was reconstituted with tris buffer.

## Transmission electron microscopy

The TEM protocol described by Dorr et al. [32] was followed, with some modification. In brief, MtM or MtM-SMALPs (10 μL containing 10 μg protein) was adsorbed (2 min) on a 100-mesh carbon film-coated copper grid (Ted Pella Inc., USA). After two washings with water, the grid was negatively stained (45 sec) with uranyl acetate (Bio Rad A-2312/3, 5 μL of 2% solution) and air-dried. Images were acquired on a Gatan Orius Model 832 camera (bundled with licensed Gatan Microscopy Suit software version 3.30.2016.0), installed on the Jeol transmission electron microscope (JEM 1400 Plus, operating at 80 kV) in the Department of Pathology, SGPGIMS, Lucknow.

## SDS-PAGE and western blotting

SDS-PAGE was performed on 12.5% resolving gel in a mini Protean-II cell (Bio Rad, USA). As only ~50% MtM proteins were solubilised by SMA, this ratio was maintained during sample loadings. The resolved antigens were either stained with Coomassie Brilliant Blue R250 or electroblotted onto nitrocellulose paper in a mini trans-blot cell (Bio Rad). For comparisons by western blotting, both the antigens (MtM and MtM-SMALP) and molecular weight markers were run side-by-side in the same gel. After electroblotting, the area corresponding to each antigen was cut into strips. The nitrocellulose paper strips containing blotted antigens were probed with monoclonal or polyclonal antibodies to Mtb antigens as follows. Strips were

blocked (2 h, RT) with 2% BSA-TBS-T (2% bovine serum albumin in 50 mM Tris, 100 mM NaCl, 0.05% Tween 20; pH 7.4) and incubated (2 h, RT) with anti-LAM mAbs (provided by WHO-TDR and Prof. Juraj Ivanyi) or sera from healthcare workers diluted (1:50) in 1% BSA-TBS-T. After extensive washing (5 times) with TBS-T, the strips were incubated (2 h, RT) with alkaline phosphatase-conjugated antibody to mouse Ig (Sigma, A0162) or peroxidase-conjugated antibody to human IgG (Sigma, A0170), both diluted (1:2000) in 1% BSA-TBS-T. After another round of washings, strips were incubated with the substrate solution (Sigma BCIP/NBT B5655 for alkaline phosphatase, and BioRad Opti-4CN substrate Kit 170–8235 for peroxidise). The colour reaction was stopped by washing with water. Some of the NCP strips were probed for the detection of Con A-binding antigens as follows [33]. Blocked strips were incubated (overnight, RT) with 5 μg/mL (in 1% BSA-TBS-T) peroxidase-conjugated Con A (Sigma, L6397). Later, the strips were washed with TBS-T and incubated with peroxidase substrate as above. Gel and blot images were acquired on ImageQuant LAS 500 (GE Healthcare) and each TIFF image was uniformly processed, if needed, for brightness and contrast using Adobe Photoshop CS6.

## Chemical cross-linking of membrane proteins

The membrane-permeable non-cleavable cross-linker disuccinimidyl suberate (DSS, Thermo 21655), which contains an amine-reactive N-hydroxysuccinimide ester at each end of the 8-carbon spacer arm, was used for chemical cross-linking of MtM proteins using the supplier's protocol. In brief, aliquots of MtM (500 μg protein in 90 μL PBS) were mixed with 10 μL of serially diluted DSS (in DMSO) to get an effective concentration of 1.25, 0.62, 0.31 and 0.15 mM. Tubes were incubated (30 min, RT) with occasional tapping. To stop the reaction, 100 μL tris buffer (100 mM, pH 7) was added to each tube and incubated at RT for 15 min.

## ELISA

A previously described method [8] was used with some modification. MtM (50 μg protein/mL) or MtM-SMALP (25 μg protein/mL, considering ~50% solubilisation of MtM proteins by SMA) diluted in coating buffer (50 mM carbonate, pH 9·5), or coating buffer alone was dispensed (50 μL/well) in U-bottom ELISA plates (Nunc Maxisorp) and incubated overnight at 4˚C. After washing with TBS-T, plates were blocked with 2% BSA-TBS-T. Later, the plates were incubated with either a set of mAbs against Mtb membrane-associated antigens or with the donor (healthcare worker) sera. The mAbs (1:50) or sera (1:500) were diluted in 1% BSA-TBS-T and dispensed (50 μL/well, in duplicate) in the antigen- and buffer-coated wells. After incubation (90 min, 37˚C), the plates were washed with TBS-T and re-incubated (90 min, 37˚C) with alkaline phosphatase-conjugated antibody to mouse Ig or peroxidase-conjugated antibody to human IgG, diluted (1:4000) in 1% BSA-TBS-T (50 μL/well). Plates were finally washed with TBS-T and the substrate solutions- pNPP (pNPP tablet set, Sigma N1891) for alkaline phosphatase and TMB (BD OptEIA reagent set, 555214) for peroxidase were added (50 μL/well) and incubated (20 min, RT, in dark). Reactions were stopped by adding (50 μL/well) 3N NaOH (for alkaline phosphatase) or 7% $H_2SO_4$ (for peroxidase). ODs were read, respectively, at 405 or 450 nm in a plate reader. For each antibody, the difference in mean OD of antigen- and buffer-coated wells was expressed as ΔOD.

## T cell proliferation and cytokine assays

For T cell proliferation assays, a previously described method [8] was used. In brief, blood samples (diluted 1:10 in RPMI medium) were dispensed (1 mL/well) in 24-well culture plates and incubated with test antigens or controls (culture medium and SMA as negative, and PHA

as positive control) for five days in a $CO_2$ incubator. Afterwards, 0.5 mL culture supernatant was collected from each well and stored at -20˚C for cytokine assays (described below). The cells were harvested, centrifuged and re-suspended in PBS. Subsequently, cells were incubated with fluorescent (APC-conjugated) anti-CD3 antibody and RBCs were lysed with the lysis buffer. Leukocytes obtained after centrifugation were washed with PBS and fixed with 2% para-formaldehyde. Cells were again washed with PBS containing 0.05% BSA (PBS-BSA) and permeabilized with 0.2% TX-100. After another wash with PBS-BSA, cells were incubated with the fluorescent (PE) anti-Ki67 antibody. Cells were finally washed with PBS-BSA and re-suspended in PBS. Data on $10^5$ cells in 'lymphocyte gate' was acquired on a FACS Canto-II flow cytometer (BD) and analysed using FlowJo™ software [34].

Assays for the cytokines interferon gamma (IFN-γ) and tumour necrosis factor-alpha (TNF-α) in culture supernatants were performed by sandwich ELISA using the respective kits (human IFN-γ, BD 555142; human TNF-α, BD 555212) and following the kit protocols.

### Statistical analysis

Differences between two datasets were computed by either Wilcoxon matched pair signed rank test (for paired data) or Mann-Whitney test (for unpaired data). P values < 0.05 were considered as significant. All statistical analyses were performed by using GraphPad Prism 7 software.

## Results

### Yields of Mtb membrane proteins by various processes

The average yield (from 3 batches) of MtM was 10 mg protein per gram wet weight of the culture-grown Mtb, which was comparable with that seen in our earlier studies [7]. The yields of MtM proteins solubilised by various membrane protein extraction methods are shown in Table 1. The best yield (52% of the input) was obtained after extraction with 2.5% SMA for 2 h in a buffer devoid of NaCl, hence all subsequent experiments were conducted with MtM-SMALPs prepared in this manner. Addition of NaCl to the buffer reduced the yield to 23%. An overnight extraction with SMA also resulted in a lower yield (36%).

Since TX-100 has been widely used as a solubilising agent for the analysis of Mtb membrane proteome [35] we also compared the protein yields of SMA and TX-100 extracts. Whereas nearly 40% of MtM proteins were solubilised by TX-100 (Table 1), nearly 30% of the TX-100 insoluble proteins could be solubilised upon re-extraction with SMA.

**Table 1. Processed and recovered MtM proteins.**

| Total processed protein in mg (sample type)[a] | Total recovered protein in mg (method) | Soluble protein in mg | Insoluble protein in mg | Soluble protein (% of total recovered) |
|---|---|---|---|---|
| 10.0 (MtM) | 7.9 (SMA, 2h) | 4.1 | 3.8 | 51.9 |
| 7.5 (MtM) | 4.4 (SMA with NaCl, 2h) | 1.0 | 3.4 | 22.7 |
| 4.5 (MtM) | 2.8 (SMA, O/N) | 1.0 | 1.8 | 35.7 |
| 8.0 (MtM) | 6.8 (TX-100) | 2.6 | 4.2 | 38.2 |
| 3.4 (TX-100 insoluble MtM) | 2.8 (SMA) | 0.8 | 2.0 | 28.6 |
| Filtration through 100 kDa cutoff filter: | | | | |
| 0.36 (MtM-SMALPs) | 0.24 | 0.03 (<100 kDa) | 0.21 (>100 kDa) | 12.5 (<100 kDa) |

[a] Same batch of MtM was used for all experiments.

An important concern was whether the MtM-SMALP suspension also contained some free (unbound to SMALP) MtM proteins. To address this, we passed the suspension through a 100 kDa cut-off filter. Only about 10% of the loaded proteins eluted into the filtrate (Table 1) suggesting that the MtM-SMALP preparation was largely free from unbound MtM proteins.

## Morphological features of MtM and MtM-SMALPs

When visualised by transmission electron microscopy, MtM-SMALPs appeared as nanoparticles of widely-ranging shapes and sizes (Fig 1A). Interestingly, apart from the expected discoid

**A**

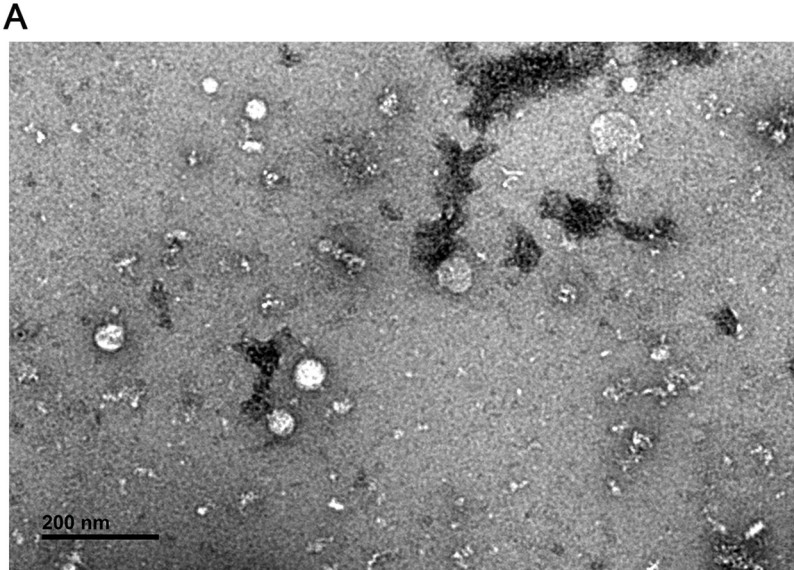

**B**

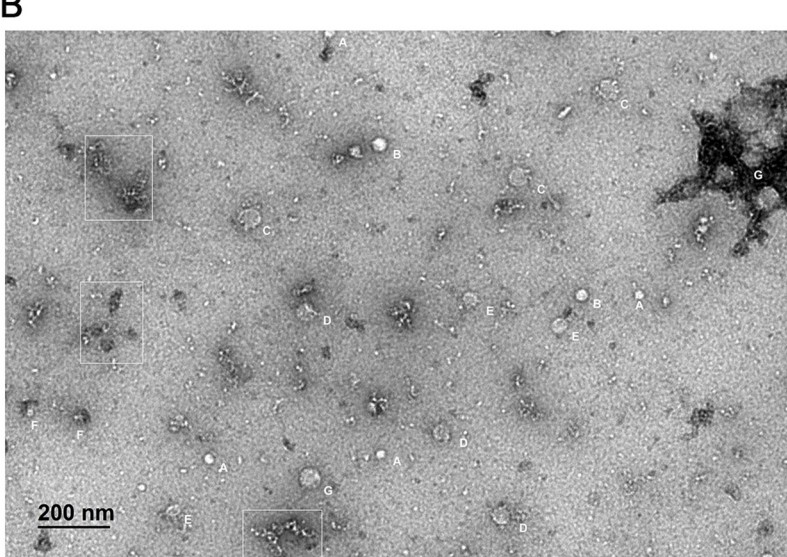

**Fig 1. TEM images of MtM-SMALPs.** A: Image at a higher magnification showing wide heterogeneity in the size and shape of SMALPs. B: Image at a lower magnification showing clusters (A-G) of two or more particles having comparable shapes or sizes. Particles within the boxed areas show 'worm-like' shapes as reported previously for SMALPs containing bacteriorhodopsin [18].

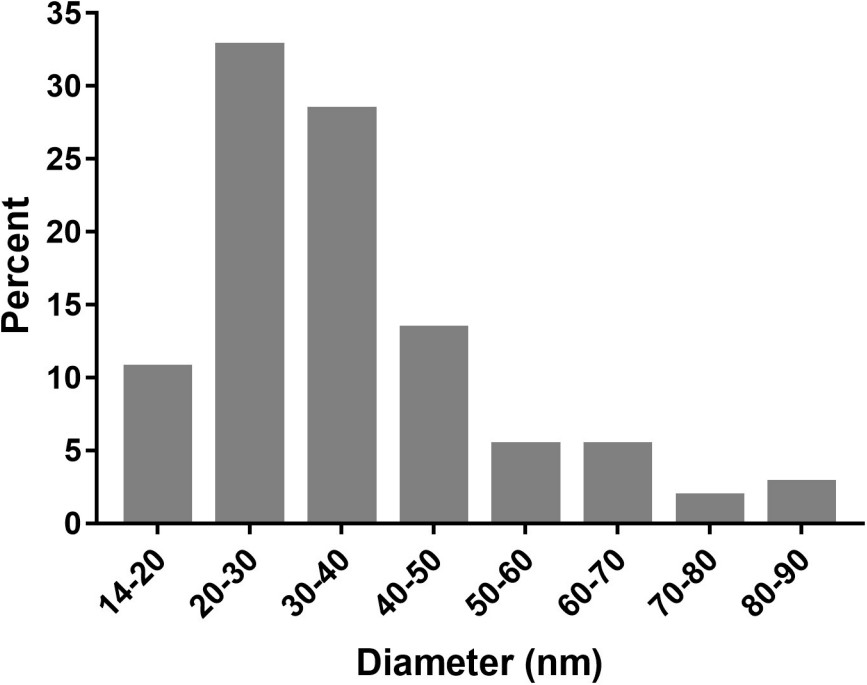

**Fig 2. Size distribution of the discoid MtM-SMALPs.**

shapes, some of the SMALPs adopted 'worm-like' shapes (Fig 1B), similar to those reported for SMALPs formed with bacteriorhodopsin [18]. Moreover, these variations did not appear as 'random'. Upon examination, the particles appeared to fall into clusters of discrete shapes and sizes (Fig 1B).

We also estimated the heterogeneity of discoid MtM-SMALPs by measuring the diameters of 113 nanoparticles (S1 Fig and S1 Table). As shown in Fig 2, majority of the particles (~61%) had a diameter of 20–40 nm. The medium frequency clusters were of particles having diameters of 40–50 nm (~13%) and 14–20 nm (~11%). However, a subset of particles (~15%) exhibited diameters in the range of 50–90 nm.

The discoid morphology of MtM-SMALPs was preserved even after lyophilisation, although the particles appeared less heterogeneous than their un-lyophilised counterparts. The particles also deposited more rapidly and densely on the copper grid (S2 Fig).

The MtM also formed vesicles of 15–25 nm diameters which coalesced in the aqueous medium owing to their hydrophobic/lipophilic propensity (S3 Fig).

## Protein profiles of MtM and MtM-SMALPs

SDS-PAGE profiles of Mtb cytosol, MtM, MtM-SMALPs and SMA-insoluble MtM are shown in Fig 3A. Although most MtM proteins were insoluble in SMA, MtM-SMALPs were found rich in proteins having a subunit molecular weight range of 40–70 kDa. Remarkably, the protein profiles of un-lyophilised and lyophilised MtM-SMALPs were identical suggesting that there was no apparent protein loss due to lyophilisation.

The profile of TX-100-soluble MtM proteins was different from that of MtM-SMALPs, although a large proportion of MtM still remained insoluble (Fig 3B). The subset of TX-100-insoluble proteins which could be solubilised upon re-extraction with SMA was

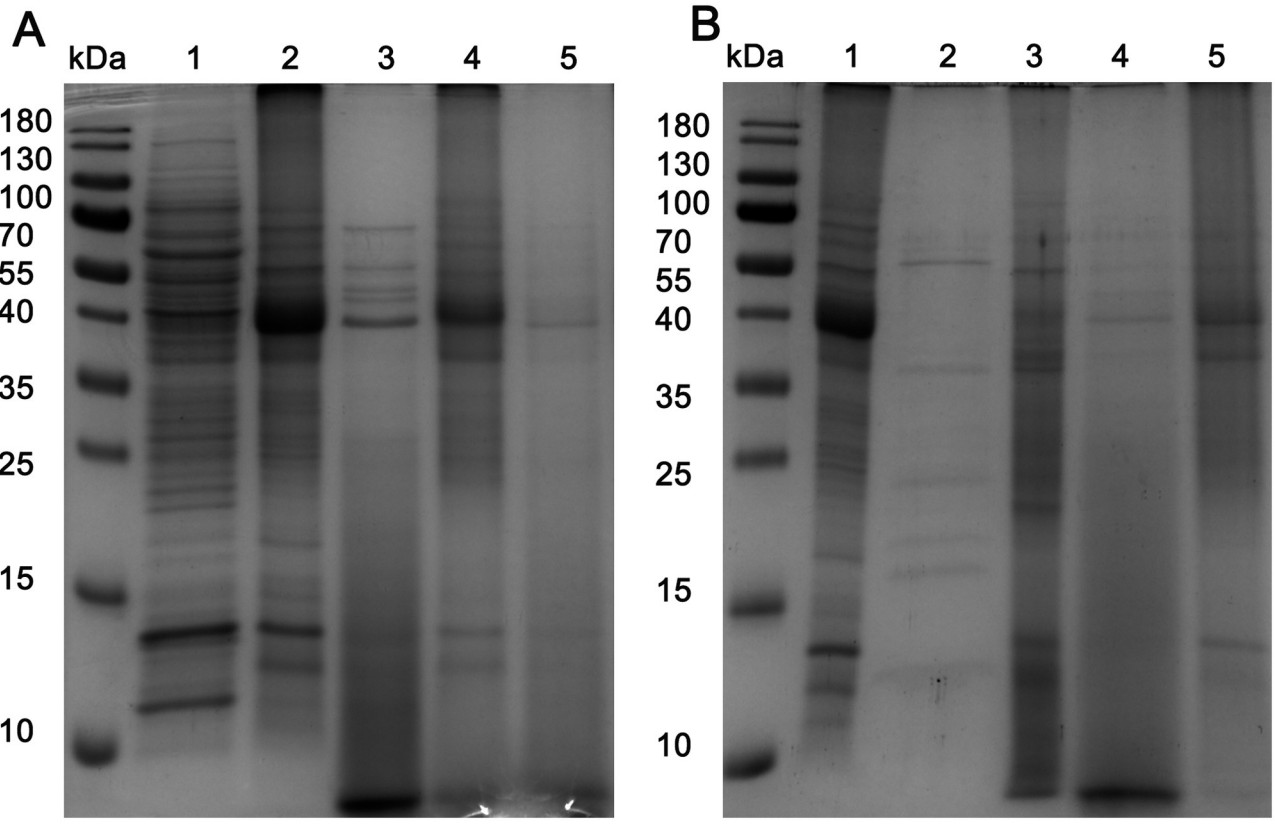

**Fig 3. SDS-PAGE profiles of MtM proteins following extraction with 2.5% SMA (panel A) or 2% TX-100 (panel B).** Lanes in panel A: 1, Mtb cytosol; 2, MtM; 3, MtM-SMALPs; 4, SMA-insoluble MtM and 5, lyophilized MtM-SMALPs. Lanes in panel B: 1, MtM; 2, TX-100-soluble MtM; 3, TX-100-insoluble MtM; 4, SMA-soluble fraction of TX-100-insoluble MtM and 5, SMA-insoluble fraction of TX-100-insoluble MtM. The TX-100-insoluble proteins which were solubilised by SMA appeared to be the same as those in MtM-SMALPs (panel A). The broad diffuse bands at the bottom of SMALP lanes (lane 3 in panel A and lane 4 in panel B) are of free SMA, which is released from SMALPs during sample preparation and gets stained with Coomassie Blue [14].

apparently the same as the population that had been solubilised directly from MtM. This observation indicated the precision with which SMA could solubilise the membrane proteins.

Many vital cellular processes are carried out by protein complexes and proteins which form such complexes are amenable to chemical cross-linking. To gain insight into the MtM proteins which might exist as macromolecular complexes, we treated MtM with a membrane-permeable, non-cleavable cross-linker disuccinimidyl suberate (DSS). SDS-PAGE profiles of DSS-treated and untreated MtM are shown in Fig 4. DSS at a concentration of 0.15–0.3 mM apparently cross-linked the proteins in the 40–70 kDa mass range (box B), which overlapped with that of the MtM-SMALP proteins. Further, the cross-linking appeared confined to only this set of proteins since the other proteins (e.g., in box C) remained unaffected. The use of higher DSS concentrations led to a more generalised (or non-specific) cross-linking [36] as evidenced by a gradual disappearance of protein bands in box C (as well as other gel areas). Concomitantly, a broad diffuse band appeared at the top of the lanes containing cross-linked proteins (box A) suggesting the formation of large protein complexes [36]. While a dark broad band was seen in lane 2 (box A), the corresponding area in lane 3 was lighter, supporting that the use of high DSS concentrations may have led to an excessive cross-linking with the complexes being too large to enter the resolving gel.

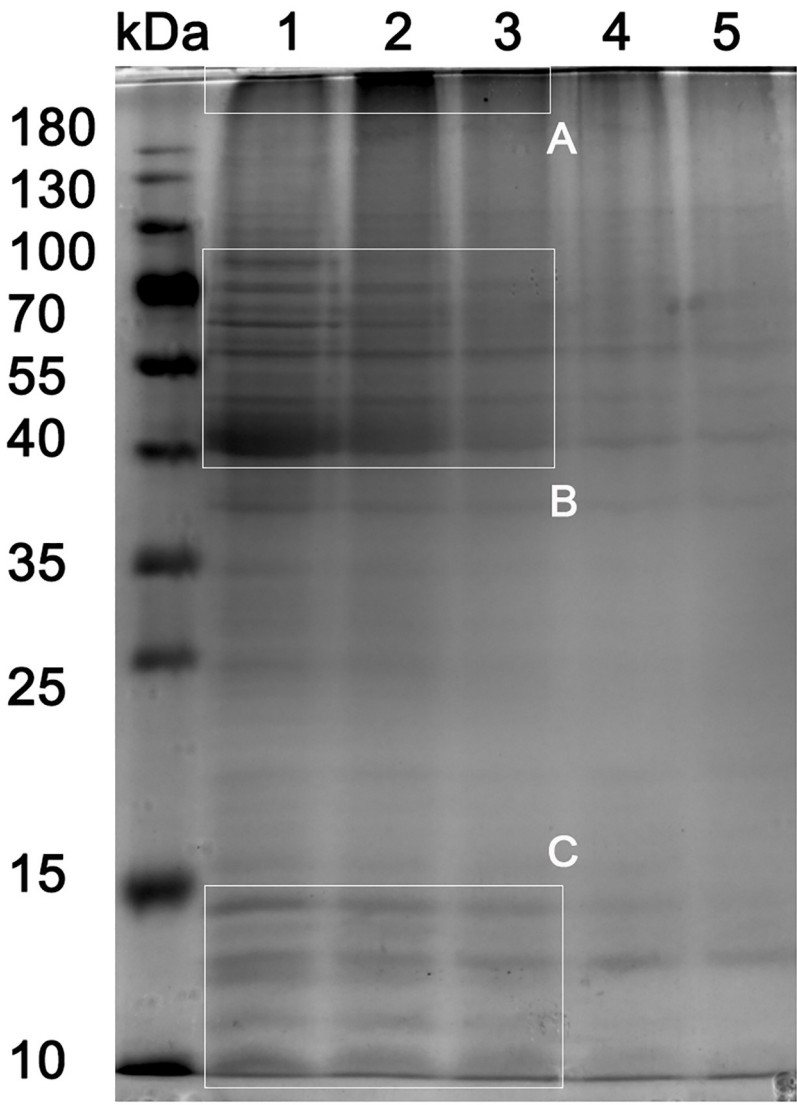

**Fig 4. Chemical cross-linking of MtM proteins visualised by SDS-PAGE.** Profiles of lanes 1–5 are as follows: 1, MtM; 2, MtM treated with 0.15 mM DSS; 3, MtM treated with 0.31 mM DSS; 4, MtM treated with 0.63 mM DSS and 5, MtM treated with 1.25 mM DSS. Box B shows the profile of MtM proteins (lane 1) which were cross-linked (lanes 2–3) by 0.15–0.31 mM DSS, as opposed to the proteins (box C, lanes 2–3) which were not cross-linked. Further increases in DSS concentration (lanes 4 and 5) led to a 'non-specific' cross-linking of proteins [36] in box C as well as other gel areas. Lane 2 in box A shows enhanced protein staining, most probably due to the accumulation of protein complexes formed by cross-linking. Corresponding area in lanes 3–5 gradually lost the staining intensity despite enhanced cross-linking. In this case, the complexes may have been too large to enter the gel.

## Immunochemical characterisation of MtM and MtM-SMALPs

To determine whether MtM represented the plasma membrane and MtM-SMALPs were rich in membrane proteins, we performed ELISA analyses with mAbs recognizing six plasma membrane-associated proteins and a plasma membrane-associated lipoglycan- lipoarabinomannan (LAM). The distinctive features of these mAbs and corresponding antigens are summarised in Table 2. PRA is a membrane protein having α-helical transmembrane domains, and PstS1 and LpqH are lipoproteins bearing triacylated 'lipobox' cysteines for attachment to the plasma membrane. Ag85 (FbpA) is a secretory protein bearing Tat motif which also associates it with

**Table 2. Definition of monoclonal antibodies used for characterisation of MtM and MtM-SMALPs.**

| WHO[a] and/or [contributor's code] for mAb | mAb class | Antigen | Gene[c] | Protein length (aa) | TM/ lipoprotein/other membrane anchors[c] |
|---|---|---|---|---|---|
| IT59 [F67-1[b]] | IgG1 | PRA (Proline Rich Antigen) | Rv1078 | 240 | Three TM domains (aa 98–118, 142–162 and 203–223) |
| IT66 [C38.D1] | IgG1 | PstS1 | Rv0934 | 374 | N-terminal membrane lipid attachment site (aa 1–24) |
| IT19 [TB23] | IgG1 | LpqH | Rv3763 | 159 | N-terminal membrane lipid attachment site (aa 1–22) |
| IT48 [HYT2] | IgG3 | Ag85 | Rv3804c | 338 | Tat-type signal (aa 1–43) for export through inner membrane |
| IT1 [F23-49] | IgG2a | Acr | Rv2031c | 144 | None (peripheral membrane protein) |
| IT63 [F86-2] | IgG1 | 35kd_ag (PspA) | Rv2744c | 270 | None (peripheral membrane protein) |
| [ML09[b]] | IgG3 | LAM[d] | - | - | Phosphatidyl-myo-inositol mannoside (PIM)-based lipid anchors[e] |
| [ML34[b]] | IgM | | | | |

[a] Khanolkar-Young et al. [37].

[b] Antibody raised against *M. leprae* but also cross-reacts with Mtb antigen (Thole et al. [38], Ivanyi, et al. [39]).

[c] https://www.uniprot.org

[d] ML09 and ML34 bind to non-overlapping epitopes on LAM [39].

[e] Batt et al. [40].

the plasma membrane. The remaining two proteins, Acr (HspX) and 35kd_ag (PspA) are peripheral membrane proteins which are rich in sequences of hydrophobic amino acids [13].

As shown in Fig 5A, all mAbs bound to MtM with high intensity signals ($\Delta$OD = 0.26–1.19) indicating high *in situ* quantities of corresponding antigens. However, the binding intensities for MtM and MtM-SMALPs were disproportionate. MtM-SMALPs showed highest affinity for PRA ($\Delta$OD for MtM-SMALP = 42% of the $\Delta$OD for MtM) whereas PstS1, LpqH, Ag85 and LAM partitioned into MtM-SMALPs with lower affinities ($\Delta$OD = 13–35% of $\Delta$OD for MtM). Conspicuously, Acr and 35kd_ag were nearly completely excluded ($\Delta$OD for MtM-SMALPs $\leq$ 5% of $\Delta$OD for MtM). These results suggested that MtM comprises the plasma membrane and SMA may preferentially extract the proteins with $\alpha$-helical trans-membrane domains. In addition, it may also extract, to a variable extent, the membrane-associated proteins or non-proteinaceous moieties such as lipoglycans. However, peripheral membrane proteins are least likely to be extracted by SMA.

Since LAM is a molecularly heterogeneous and immunologically important constituent of Mtb plasma membrane [40], we performed immunoblotting with corresponding mAbs (ML09 and ML34) to look for the presence of its various isoforms within MtM and MtM-SMALPs. Multiple LAM moieties were recognised in MtM (Fig 5B, lanes 1 and 3) which was not surprising since several chemically-diverse forms of LAM coexist [40]. Consistent with the ELISA results, MtM-SMALPs showed only a few faint LAM bands upon staining with ML34 (Fig 5B, lane 2). On the other hand, despite showing a higher binding ($\Delta$OD) to MtM in ELISA, ML09 produced a weaker staining (compared with ML34) of LAM suggesting poor accessibility of the corresponding epitope after SDS-PAGE and electro-blotting.

Protein glycosylation plays an important immunomodulatory role and a large number of MtM-associated proteins are known to be glycosylated. We therefore investigated whether the glycosylated proteins would also partition into MtM-SMALPs. Binding of Concanavalin A (Con A) to the electroblotted MtM (Fig 5B, lane 5) and MtM-SMALPs (lane 6) showed that the high-mannose glycoproteins were mostly confined to MtM as they did not apparently partition into MtM-SMALPs. Interestingly, LAM moieties below 35 kDa (comprising the

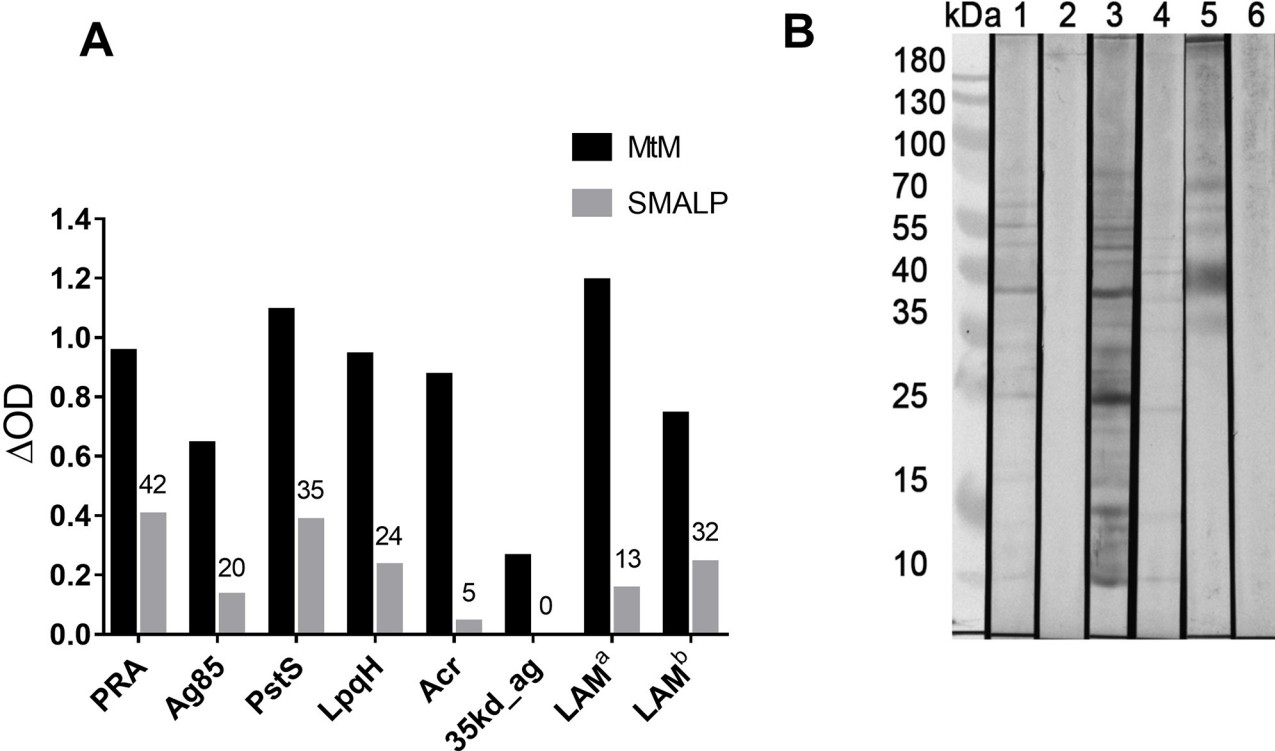

**Fig 5. Immunochemical characterisation of MtM and MtM-SMALPs.** Panel A shows intensity of binding (average of two ΔOD values measured by ELISA) of mAbs recognizing seven plasma membrane-associated antigens of Mtb (six proteins and a lipoglycan-LAM) to MtM and MtM-SMALPs. Numbers on top of SMALP bars denote proportion (%) of the corresponding binding (ΔOD) to MtM. For characterisation of LAM, two mAbs (ML09 and ML34), which bind to non-overlapping epitopes (LAM[a] and LAM[b]), were used. Panel B shows results of western blotting with ML09 (strips 1 and 2) ML34 (strips 3 and 4), and Con A (strips 5 and 6). The electroblotted antigens were either MtM (strips 1, 3 and 5) or MtM-SMALPs (strips 2, 4 and 6).

membrane anchor PIMs) also did not bind to Con A suggesting that the mannose residues in this case were inaccessible for lectin binding.

## Preferential induction of cell-mediated immune responses by MtM-SMALPs

One of the objectives of studying Mtb membrane proteins is to eventually exploit them for the development of a vaccine or as the basis for a diagnostic test for TB. We therefore determined the cell-mediated and humoral (antibody) immune responses of the study subjects (donors) against MtM and MtM-SMALPs. All donors were positive for the tuberculin skin test, suggesting the presence of a latent TB infection.

As shown in Fig 6A, the T cell proliferative responses to MtM and MtM-SMALPs were comparable (median responses, 0.60 and 0.65%; P = 0.2813) and so were the responses (0.06 and 0.1%, P = 0.6250) to negative controls (culture medium and SMA). Responses to MtM-SMALPs before and after lyophilisation was tested in one donor and found to be comparable (Fig 6A, inset). Reflecting the cell-proliferative responses, both MtM and MtM-SMALPs also induced similar levels of IFN-γ, a cytokine produced mainly by the T cells (Fig 6B). However, a remarkable difference was seen in the levels of TNF-α, a cytokine produced mainly by the monocyte-macrophages (apart from some other blood cells). Although none of the six donors produced TNF-α in response to MtM (all levels were equal to the negative controls),

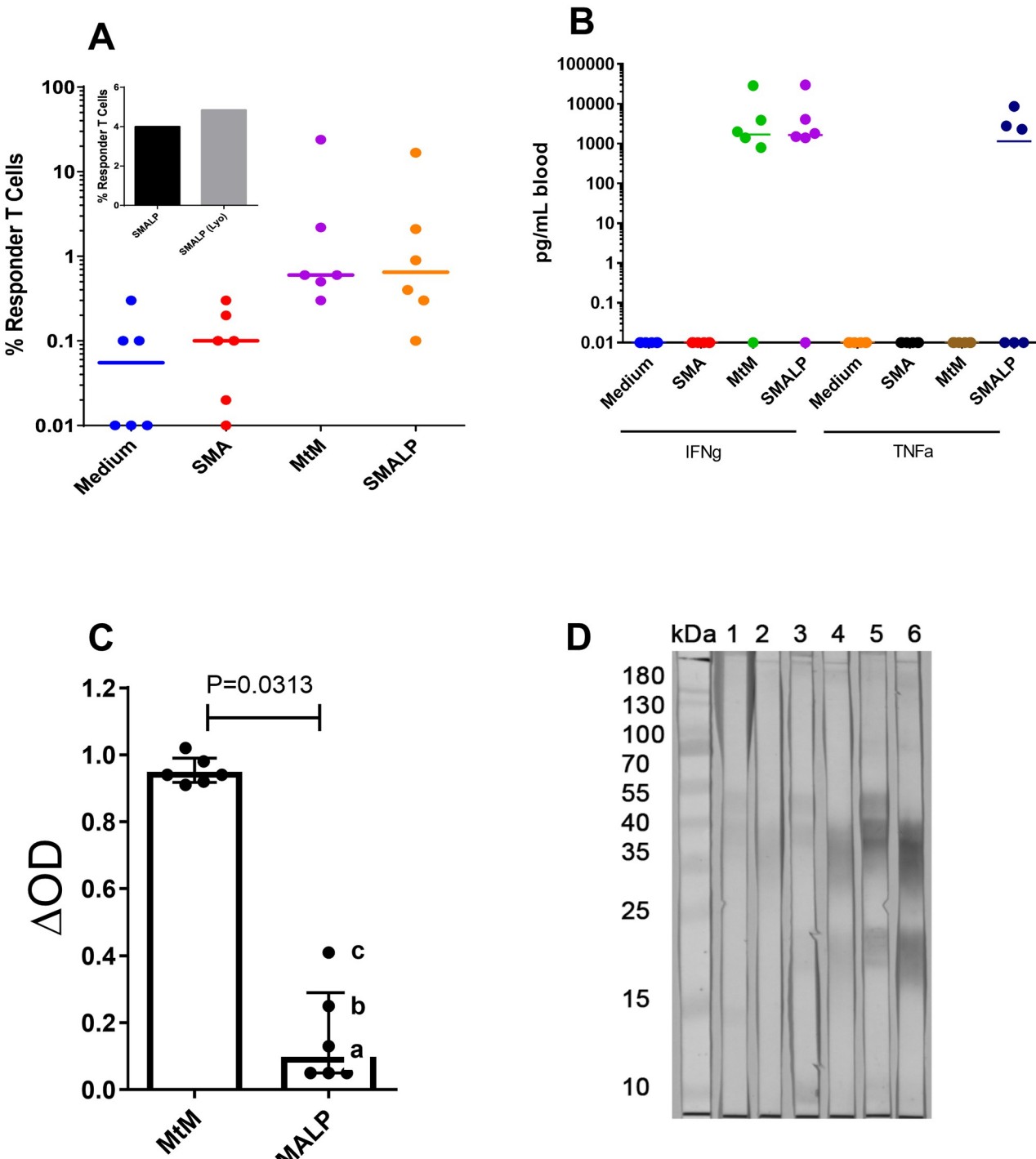

**Fig 6. Cell-mediated and humoral immune responses to MtM and MtM-SMALPs in donors with latent TB infection.** Panel A shows proliferative T cell responses to both the antigens as well as negative controls (culture medium and SMA). Inset graph shows response of one donor to MtM-SMALPs before and after lyophilisation. Panel B shows concentration (expressed as pg/mL blood) of the cytokines IFN-γ and TNF-α in culture medium. Only MtM-SMALPs induced the production of TNF-α in three donors. Panel C shows the serum antibody levels (average of two ΔOD values measured by ELISA) against MtM and MtM-SMALPs. Panel D shows the results of immunoblotting with three donor sera having highest antibody levels against MtM-SMALPs (marked with letters a-c in panel C). Strips 1, 3 and 5 contained electroblotted MtM and strips 2, 4 and 6 contained MtM-SMALPs. Strips 1 and 2 were reacted with serum-a, 3 and 4 with serum-b, and 5 and 6 with serum-c.

three of them produced this cytokine upon stimulation with MtM-SMALPs. Contrary to the cell-mediated immune responses, levels of serum antibodies (ΔOD values) against MtM-SMALPs were significantly lower than those against MtM (Fig 6C). Results of immuno-blotting with sera from the three donors who had highest antibody levels against MtM-SMALPs (marked with letters a-c in Fig 6C) are shown in Fig 6D. As expected, strongest reactivity was shown by serum-c, followed by serum-b and -a. The immunoblot results with MtM are comparable with those reported in our earlier studies [23].

These results suggested that MtM-SMALPs preferentially induced the T cell and macrophage mediated immune responses in donors harbouring a latent TB infection.

## Discussion

SMA copolymers have unique and versatile membrane solubilising properties and have shown efficient solubilisation of membrane proteins from a variety of organisms. In studies comparing SMA copolymers of varying styrene to maleic acid ratios, SMA 2:1 (SMA 2000) has emerged as the best option for extraction, purification and stabilization of membrane proteins [41, 42]. These observations prompted us to use SMA 2:1 for this study. Under the optimised conditions SMA could solubilise ∼50% of MtM proteins which was comparable with the earlier reports [41, 42]. However, contrary to the expectations [14] inclusion of NaCl in the reaction mixture was detrimental to solubilisation and turbidimetry did not help monitor the solubilisation process. Nonetheless, SMA appeared to be a more efficient solubilising agent than TX-100 which has been used extensively in the study of Mtb membrane proteome [35]. SMA exhibits a distinctly different mode of action than detergents. While SMALPs retain the native lipids surrounding membrane proteins, detergent micelles strip those lipids. Thus the ability of detergents to extract proteins from membranes is dependent on their ability to solubilise the membrane lipids, some of which form detergent-insoluble 'rafts' [43]. The efficiency of protein solubilisation from any membrane depends on several factors including lipid/protein ratio and composition and packing of membrane lipids [15]. Mtb has a unique membrane comprising some unusual phospholipids, such as the phosphatidyl-myo-inositol mannosides (PIMs). Over 50% of the PIMs is PIM2 (two mannose residues on the inositol moiety) which is present exclusively in the inner leaflet of plasma membrane. The outer leaflet is mainly occupied by PIM5 and PIM6' [40, 44]. This unusual arrangement leads to tight packing of fatty acyl chains within the plasma membrane making it less fluid, more stable and less permeable. In addition, PIMs also serve as membrane anchors for the lipoglycans- lipomannan (LM) and LAM.

Under transmission electron microscopy, SMALPs prepared from native Mtb membranes displayed wide-ranging shapes and sizes (∼10–90 nm diameter) including some 'worm like' shapes described previously for the SMALPs incorporating bacteriorhodospsin [18]. Although, to our knowledge, there is no prior study showing TEM images of SMALPs prepared from whole membrane proteome, this heterogeneity in shape/size likely reflects the vast variation in membrane proteins of Mtb in terms of their molecular masses or trans-membrane helix numbers. For example, the membrane protein NuoN (Rv3158, 55.34 kDa, a NADH dehydrogenase) has 14 trans-membrane helices [12, 13]. This observation also contrasted with the near-homogeneous small size (typically ∼10 nm diameter) of SMALPs incorporating a single protein or a protein complex [15, 41] and belied the apprehension that SMALPs, owing to their small size, may not be able to accommodate large membrane proteins or membrane protein complexes. Remarkably, the variations in shape and size of MtM-SMALPs did not appear 'random' as the particles fell into clusters of discernible shapes and/or sizes. It is tempting to speculate that each of these clusters could contain membrane proteins or protein complexes of

distinct structural features. The shape/size variation in lyophilised MtM-SMALPs was less remarkable, which could be due to the after-effect of lyophilisation. The TEM image of MtM showed clusters of vesicles formed spontaneously in the aqueous medium. Formation of such vesicles, typical of a plasma membrane, was also reported by us for a 'fast-grower' mycobacterium- *M. fortuitum* [45] and by Chiaradia et al. [46] for another fast-grower, *M. smegmatis*. In the latter study, it was also shown that the mycobacterial outer membrane appears as 'unclosed' fragments suggesting that our MtM preparation represented the plasma (or inner) membrane of Mtb. Model membrane vesicles prepared from the lipids extracted from mycobacterial cell membranes could, however, be much larger in size [47].

When resolved by SDS-PAGE, MtM-SMALPs displayed a set of proteins, mostly of ~40–70 kDa weight range. Nevertheless, the existence in MtM-SMALPs of some other less abundant proteins, which could not be stained by coomassie blue, cannot be ruled out. The majority of MtM proteins that were not incorporated into SMALPs could be peripheral membrane proteins [7] which are known to be associated with the membrane lipid bilayer through electrostatic or hydrophobic interactions [25]. Alternatively, a proportion of SMA-insoluble MtM proteins could indeed be membrane proteins since their solubilisation by SMA depends on their structure and the composition of the surrounding membrane lipids [18]. Although we have not used other SMA copolymers, some of the more-recently introduced copolymers may show differing efficiency in membrane protein extraction from the Mtb membrane which has a complex and unique structure. However, this will have to be determined empirically. The subset of proteins solubilised by TX-100 was different from that solubilised by SMA, which could have resulted from the preferential solubilisation of peripheral membrane proteins by TX-100 [7, 43]. Interestingly, the set of proteins solubilised by SMA from the TX-100-insoluble MtM overlapped with that from the native MtM, suggesting that the process of solubilisation of membrane proteins by SMA was specific and robust. Lyophilisation of MtM-SMALPs apparently did not result in any loss of their constituent proteins, which makes it convenient to use the lyophilized particles for future structure/function studies. In the hands of other workers also, proteins in SMALPs have shown a remarkable stability [15].

Proteins rarely function independently. Hence, elucidation of protein-protein interactions is crucial to our understanding of the sub-cellular processes. As a step towards identification of MtM proteins which could participate in the formation of macromolecular complexes, we treated the MtM with a membrane-permeable cross-linker (DSS) which covalently cross-links the interacting proteins [36]. MtM proteins of a mass range overlapping with that of MtM-SMALP proteins (40–70 kDa) appeared amenable to cross-linking. Although the molecular identity of these proteins was not determined in this study, Zheng et al. [48] were able to identify the BCG membrane proteins involved in protein-protein interactions using a combination of blue native PAGE and LC-MS/MS. Hence, opportunities to identify the interacting membrane proteins from Mtb using SMALP-captured complexes promise to be a fruitful area for future exploration.

Some key characteristics of the MtM constituents extracted by SMA were revealed by their binding to the mAbs against seven membrane-associated antigens of Mtb. As expected, MtM-SMALPs showed highest preference for the protein (PRA) bearing transmembrane α-helices. In addition they also incorporated, although to a lesser extent, the lipoproteins PstS1, LpqH and the secreted protein Ag85 which do not bear any transmembrane helix. There are some compelling reasons for the presence of these proteins in the plasma (inner) membrane. The 'lipobox' cysteines of PstS1 and LpqH are covalently lipidated with three fatty acyl chains which anchor them to the outer face of plasma membrane. This association is believed to be further strengthened by glycosylation of both the proteins [49]. In an earlier study also, a triacylated membrane-anchored lipoprotein of *Flavobacterium* was shown to partition into

SMALPs [20]. PstS1 and LpqH have also been described as 'secreted' proteins. However, since bacterial secreted proteins are typically non-lipidated, the release of PstS1 and LpqH into the culture medium has been attributed to 'shedding' or 'shaving' rather than secretion [49]. The mycolyl transferase Ag85 (FbpA) bears a Tat-type signal motif for export across the plasma membrane [13, 50]. Hence, a proportion of Ag85 is likely to be present in the plasma membrane at any given time point. Partitioning of the MtM-associated lipopoglycan LAM into MtM-SMALPs further suggests that even non-proteinaceous moieties may become incorporated into SMALPs provided that they are anchored into the plasma membrane via lipidic tails [40]. Remarkably, the peripheral membrane proteins Acr (HspX) [51] and 35kd_ag (PspA) [52] were not incorporated into MtM-SMALPs despite being rich in hydrophobic amino acid sequences [13]. This observation, that peripheral membrane proteins are unlikely to be extracted by SMA, is of special interest since their abundant presence has persistently hindered the analysis of membrane proteins [7, 35].

The glycosylated constituents of MtM showed abundant binding to Con A, with a binding pattern similar to that reported in an earlier study [53]. Conspicuously, the low molecular mass (< 35 kDa) LAM moieties, which include LM and the membrane anchor PIMs [40, 44, 45] did not react with Con A suggesting that their mannose residues may be inaccessible to the lectin. Additionally, Con A did not stain any of the major MtM-SMALP proteins. However, glycosylation of some minor constituents, not detectable by SDS-PAGE and western blotting, cannot be ruled out. For example, both the lipoproteins PstS1 and LpqH, which were partially incorporated into the SMALPs, are glycosylated [54]. In future, the use of other lectins (apart from Con A) may provide a more comprehensive profile of the glycosylated MtM proteins. A recent proteomic study has documented a large number of membrane-associated glycoproteins in Mtb, although the majority of those could be peripheral membrane proteins [54].

Immune responses of the healthcare workers harbouring a latent TB infection revealed the distinction between MtM and MtM-SMALPs. While both induced equivalent T cell responses (in terms of cell proliferation and IFN-γ production), antibody responses to MtM were significantly higher than to MtM-SMALPs. The most important difference between two preparations was in their ability to induce the production of TNF-α, a cytokine produced mainly by macrophages and, to a lesser extent, by other blood cells [1]. While none of the six donors produced TNF-α in response to stimulation with MtM, three of them did produce this cytokine in response to MtM-SMALPs suggesting an enrichment of the TNF-α-inducing membrane constituents within SMALPs. TNF-α confers primary protection against Mtb by facilitating the formation of lung granulomas within which the bacilli are confined and eventually eliminated [1]. The SMALP-bound LAM as well as the lipoproteins PstS1 and LpqH are known to bind to 'toll-like' receptors present on the surface of macrophages and induce them to produce TNF-α [1, 40, 49]. Interestingly, as evident from the western blots, LAM epitopes recognised by the mAbs ML09 and ML34 were differently displayed in MtM and MtM-SMALPs. The ML09 epitope was displayed less prominently than ML34 epitope, despite showing more intense binding to MtM in ELISA. This could have important implications for the immuno-modulatory properties of SMALP-bound LAM and its precursors or isoforms [55]. Altogether, these observations suggest that MtM-SMALP may be an attractive resource for designing a vaccine against TB. SMALPs have already been found suitable for clinical applications [27, 28].

Our study has some limitations. We have not conclusively shown that the MtM or MtM-SMALP preparations are free from constituents of the Mtb 'outer membrane'. The mycobacterial outer membrane comprises a non-conventional lipid bilayer wherein inner leaflet is made of mycolic acids and the outer leaflet is made of free (unbound) lipids [40, 47]. Covalent linkages between peptidoglycan, arabinogalactan and mycolic acids within the cell

envelope make it difficult to separate the inner and outer membranes [56]. However, while the SMA polymers can penetrate the membrane phospholipid bilayer (as is the case with inner membrane) to produce SMALPs, they may not be able to penetrate the peptidoglycan-arabino-galactan layer to which the mycolic acids are covalently linked. Notwithstanding this, certain structural features also differentiate the inner from outer membrane proteins. Inner membrane proteins are generally characterized by the presence of transmembrane α-helical domains, although certain lipoproteins may also be anchored to the inner membrane by acylation of their N-terminal cysteines with long-chain fatty acids [25]. Outer membrane proteins, on the other hand, are typically β-barrel proteins [57]. Importantly, they neither have transmembrane domains nor are they acylated with fatty acids [56]. Hence, although it is possible for certain outer membrane proteins to be associated with the MtM (as peripheral membrane proteins), the possibility of their association with MtM-SMALPs appears less likely. Another limitation of this study is that we have determined the immune responses to MtM-SMALPs in only a small number of subjects harbouring latent TB infection. Hence, although the results are intriguing, it will be important and informative to extend these findings in a larger group of subjects representing various stages of the Mtb infection [9].

What could be the way forward with SMALPs which contain native membrane proteins or membrane protein complexes of wide-ranging shapes and sizes? To begin with, it may be possible to isolate various particle clusters in the anticipation that each type will represent a unique membrane protein or membrane protein complex. Apart from the currently available tools [58], the technology for separation of nanoparticles on the basis of shape and size is evolving rapidly [59]. The next obvious step is to unravel the protein and lipid composition of MtM-SMALPs through appropriate proteomic and lipidomic approaches. The analysis of lipids assumes special significance in the light of cytokine responses generated by MtM-SMALPs. The Mtb cell-envelope lipids are known to bind CD1 molecules which can induce human T cells to secrete various cytokines including IFN-γ and TNF-α [60]. Hence, a part of the cytokine responses induced by MtM-SMALPs may be attributable to these lipids. Finally, the preferential induction of human T cell and macrophage responses by MtM-SMALPs could potentially be harnessed for the development of a subunit vaccine against TB.

## Supporting information

**S1 Raw images. Original unprocessed images for Figs 1A, 1B, 3A, 3B, 4, 5B and 6D.**
(PDF)

**S1 Fig. (A-F)**. Measurement of diameter of discoid MtM-SMALPs in TEM images.
(PDF)

**S2 Fig. TEM image of MtM-SMALPs after lyophilisation and reconstitution with water.**
(PDF)

**S3 Fig. TEM image of Mtb plasma membrane vesicles.**
(PDF)

**S1 Table. Diameters (nm) of discoid nanoparticles visualised in S1A–S1F Fig.**
(PDF)

**S2 Table. Raw data for Figs 5A and 6A–6C.**
(PDF)

## Acknowledgments

SS was an Emeritus Scientist of Indian Council of Medical Research. The panel of anti-mycobacterial monoclonal antibodies used in this study was provided by IMMYC Monoclonal Antibody Bank of WHO-TDR (courtesy of Dr TM Shinnick, CDC Atlanta, USA) and by Prof Juraj Ivanyi (King's College, London, UK).

## Author Contributions

**Conceptualization:** Sudhir Sinha, Barbara Imperiali.

**Data curation:** Sudhir Sinha.

**Formal analysis:** Sudhir Sinha, Vinita Agrawal.

**Funding acquisition:** Amita Aggarwal.

**Investigation:** Sudhir Sinha, Shashikant Kumar, Komal Singh, Fareha Umam, Vinita Agrawal.

**Methodology:** Sudhir Sinha, Shashikant Kumar, Komal Singh, Vinita Agrawal, Barbara Imperiali.

**Project administration:** Sudhir Sinha, Amita Aggarwal.

**Resources:** Sudhir Sinha, Vinita Agrawal, Amita Aggarwal, Barbara Imperiali.

**Supervision:** Sudhir Sinha, Amita Aggarwal, Barbara Imperiali.

**Validation:** Sudhir Sinha, Barbara Imperiali.

**Visualization:** Sudhir Sinha, Vinita Agrawal.

**Writing – original draft:** Sudhir Sinha.

**Writing – review & editing:** Amita Aggarwal, Barbara Imperiali.

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
