## [Decision Letter · Decision Letter 0]

9 Nov 2022

PONE-D-22-27916Immunochemical characterisation of styrene maleic acid lipid particles prepared from Mycobacterium tuberculosis plasma membranePLOS ONE

Dear Dr. Sinha,

Thank you for submitting your manuscript to PLOS ONE. After careful consideration, we feel that it has merit but does not fully meet PLOS ONE’s publication criteria as it currently stands. Therefore, we invite you to submit a revised version of the manuscript that addresses the points raised during the review process.

Both reviewers had concerns with figures, in particular TEM images and the clarity of the discoid shapes and how they are altered with SMALPs preparation methods. also please address the gel image quality issues pointed out by reviewer 2. Reviewer 2 made note of the extensive use of abbreviations and the difficulty it creates for the reader. Please edit and make judicious use of abbreviations.  Please address issues of clarity and rationale in the text identified by the reviewers. 

We look forward to receiving your revised manuscript.

Kind regards,

Christopher W Reid, Ph.D

Academic Editor

PLOS ONE

Reviewers' comments:

Reviewer's Responses to Questions

**Comments to the Author**

1. Is the manuscript technically sound, and do the data support the conclusions?

Reviewer #1: Partly

Reviewer #2: Partly

2. Has the statistical analysis been performed appropriately and rigorously? 

Reviewer #1: Yes

Reviewer #2: I Don't Know

3. Have the authors made all data underlying the findings in their manuscript fully available?

Reviewer #1: Yes

Reviewer #2: Yes

4. Is the manuscript presented in an intelligible fashion and written in standard English?

Reviewer #1: No

Reviewer #2: Yes

5. Review Comments to the Author

Reviewer #1: The authors report the use of SMA in extracting MPs from Mtb membranes and their potential use in future as diagnostic or vaccine candidates. The manuscript is well written, but suffers from absence of critical information which can support the statements made by the authors. The specific points below, need attention.

1. In the TEM images (e.g., 1A and S3), the discoid shapes are not very clear. Can the authors estimate the %s of various shapes and how they are alters with SMALPs preparation methods like concentration, ratio, time etc.

2. In TEM images such as Fig. S4, the authors show very small sizes of the plasma membrane vesicles. Why is that? Other published reports such as Biophysical Journal (Volume 118, Issue 6, 2020, Pages 1279-1291) show bigger sizes of the inner plasma membrane vesicles.

3. The authors comment that they cannot rule out the contamination by outer Mtb membrane lipids. They can just semi-qualitatively confirm this by extracting lipids from their SMALPs using standard methanol: chloroform extraction and do a standard TLC and compared the bands with published reports. This can be performed easily and added to the manuscript. Though lipidomic analysis would be best.

4. Authors should provide more information on the mechanism of SMA mediated MPs-containing SMALP formation. And in this context, explain why mainly inner membrane lipids are involved in forming the LPs and not outer membrane lipids. Also, this could help explain by TX-100 profile is different from SMALPs.

5. Also, why does SMALPs prefer helical proteins? This information should be explained in the light of their findings.

6. It is interesting to see presence of LAMs in their preparation of SMALPs. How this method is different from the reported process of extracting LAMs (See Proc. Natl. Acad. Sci. USA, 111 (2014), pp. 4958-4963).

7. Why not all donor samples were tested with lyophilized SMALPs. Only 1 sample is not enough to conclusively comment on the effect of lyophilization.

8. Why PMPs are unlikely to be extracted by SMA? This information should be provided at the relevant place.

9. Authors should show what possible TNF-a inducers are present in their SMALPs, in addition to LAM.

Reviewer #2: The authors present data describing attempts to isolate membrane proteins from Mycobacterium tuberculosis using SMA polymer. In general the paper is technically sound, with a couple possible exceptions (noted below). I have some suggestions for editing a revised manuscript.

Line 47 should reference current WHO numbers given that they have been released. I’m not convinced that ‘membrane proteins’ needs to be abbreviated (MP). This hurts readability. In fact, generally through the manuscript there is an over-reliance on abbreviations. I found myself constantly trying to remember which abbreviations meant what, especially because many are close to each other and are not 'standard'. If there is no hard cap on word limit, there is no reason to abbreviate so much. For readability, the formatting of tables could be improved. The headings of the table should at least be bolded.

Regarding the data itself, the quality of the gel images is poor. They look like they were not de-stained long enough prior to imaging, or that the running buffer was contaminated resulting in extremely high background. Many of the lanes also appear over-loaded. If these samples are still available they should be re-run and properly destained to obtain a more appropriate signal to background.

The reason for lyophilising the samples is not really explained in the manuscript. What were the authors hoping to achieve by doing that? This is sort of touched on around line 510, but should be explained when the experiment is first introduced. Otherwise the reader is left a bit confused about why that choice was made. Do the authors have any evidence that these proteins will be functional after lyophilisation? What percent of proteins can be reproducibly assayed after lyophilisation – does that number actually change with SMALP-ing?

It is not clear how many replicates are presented in figure 4? If this is only one replicate, it probably should not be included in the manuscript.

For the data around line 430. Was a SMALP-only control conducted? Do we know the response is not to the SMA polymer itself? Was the MtM sample treated the same way otherwise as the SMA such that the samples are matched for any other buffers/reagents?

In the discussion, the authors posit that being a lipoprotein is why LpqH is more abundant in SMALPs. Were other lipoproteins similarly abundant? There are many lipoproteins in M. tuberculosis. If not, then the lipidation is not the reason for better extraction into SMALPs, and this should be re-written. Is LpqH abnormally abundant?

6. PLOS authors have the option to publish the peer review history of their article (what does this mean?). If published, this will include your full peer review and any attached files.

Reviewer #1: No

Reviewer #2: No

---

## [Author Response · Author response to Decision Letter 0]

12 Dec 2022

Response to Reviewers

Reviewer #1: The authors report the use of SMA in extracting MPs from Mtb membranes and their potential use in future as diagnostic or vaccine candidates. The manuscript is well written, but suffers from absence of critical information which can support the statements made by the authors. The specific points below need attention.

1. In the TEM images (e.g., 1A and S3), the discoid shapes are not very clear. Can the authors estimate the %s of various shapes and how they are alters with SMALPs preparation methods like concentration, ratio, time etc.

Response: Thanks for these very helpful suggestions. We have replaced Figures 1A and S3 with new images. The new images offer better clarity and, as recommended, we have determined the range of diameters of the discoid MtM-SMALPs. This new data is incorporated into Figure 2, Supplementary Figure S1, Supplementary Table S2, and text (Results, lines 355-360) in the revised manuscript. The non-discoidal MtM-SMALPs were difficult to count since their shapes were vastly heterogeneous. We have performed all the experiments (including TEM) with the MtM-SMALPs which were prepared under the best solubilisation conditions (we have now clarified this in the Results, lines 312-314). However, as mentioned in the Discussion (lines 608-612), this is a work in progress as one needs to try out other SMA polymers as well.

2. In TEM images such as Fig. S4, the authors show very small sizes of the plasma membrane vesicles. Why is that? Other published reports such as Biophysical Journal (Volume 118, Issue 6, 2020, Pages 1279-1291) show bigger sizes of the inner plasma membrane vesicles.

Response: In the quoted study, lipids extracted from the inner membrane (IM) or outer membrane (OM) were reconstituted into ‘model’ membranes to generate large/giant vesicles. Our study is based on the native membrane comprising all of its constituents (lipids, proteins, carbohydrates and conjugates thereof). The size of our membrane vesicles is comparable with those prepared from the native membranes of M. fortuitum (Ref 45) or M. smegmatis (Ref 46). We have now clarified this aspect in the revised Discussion (lines 591-598). 

3. The authors comment that they cannot rule out the contamination by outer Mtb membrane lipids. They can just semi-qualitatively confirm this by extracting lipids from their SMALPs using standard methanol: chloroform extraction and do a standard TLC and compared the bands with published reports. This can be performed easily and added to the manuscript. Though lipidomic analysis would be best.

Response: The lipidomic analysis of MtM-SMALPs is indeed an attractive proposition which we would like to pursue in a future study (Discussion, lines 730-735). However, we are not sure if it will provide an answer to the query (i.e., whether a lipid moiety has come from the IM or OM) for the following reasons: (i) Many of the Mtb cell envelope lipids are shared by both the membranes and (ii) SMA may not be able to extract lipids from the OM which is a ‘non-conventional’ membrane comprising an inner leaflet of mycolic acids and outer leaflet of free lipids. Although the free lipids (their composition is still a subject of investigation) can be stripped from the cell envelope, most of the mycolic acids are covalently linked to the arabinogalactan-peptidoglycan layers of the cell envelope. In the process of SMALP formation (Introduction, 4th paragraph), the SMA polymer inserts into the phopholipid bilayer of a conventional membrane (as is the case with IM) and its aryl and meleic acid moieties interact, respectively, with the lipid acyl chains and headgroups. In view of this mechanism of action, SMA is highly unlikely to be able to insert into the layers of peptidoglycan-arabinoglycan. Furthermore, the IM lipids, owing to their saturated acyl chains, demonstrate higher membrane order. On the other hand, OM lipids show low rotational freedom due to the presence of methyl branches, cyclopropane rings, and double bonds- all of which may impede their solubilisation by SMA.

Finally, our belief that we are dealing with the IM (not OM) is supported by a study (Ref 46) in which IM was shown to form closed vesicles (similar to those seen by us) whereas OM appeared as large, unclosed fragments (Discussion, lines 594-596). 

4. Authors should provide more information on the mechanism of SMA mediated MPs-containing SMALP formation. And in this context, explain why mainly inner membrane lipids are involved in forming the LPs and not outer membrane lipids. Also, this could help explain by TX-100 profile is different from SMALPs.

Response: As suggested, we have added more information on the mechanism of SMALP formation in the revised Introduction (paragraphs 3 and 4). In the light of this mechanism, we have also explained why outer membrane lipids may not be solubilised by SMA (Discussion, lines 705-708) and why TX-100 profile is different from SMALPs (Discussion, lines 558-561). However, the mechanism of SMA action is yet to be fully elucidated. There is some clarity on the solubilisation of lipids by SMA from ‘model’ membranes, but not so much from native membranes which was the subject of our study. 

5. Also, why does SMALPs prefer helical proteins? This information should be explained in the light of their findings.

Response: As mentioned above, we have now elaborated the mechanism of SMA action in the revised Introduction. Accordingly, SMA acts like a ‘cookie cutter’ to generate nanoparticles from the membrane lipid bilayer. Being embedded in the bilayer, the alpha helical trans-membrane proteins become a part of these nanoparticles.

6. It is interesting to see presence of LAMs in their preparation of SMALPs. How this method is different from the reported process of extracting LAMs (See Proc. Natl. Acad. Sci. USA, 111 (2014), pp. 4958-4963).

Response: According to the mechanism now elaborated in the Introduction, SMA did not specifically extract LAM. It extracted patches of the membrane lipid bilayer which happened to contain LAM anchored into the bilayer through its PIM tails (Ref 40). The quoted paper (Ref 44) describes the method for LAM purification from delipidated mycobacterial cells/ cell walls.

7. Why not all donor samples were tested with lyophilized SMALPs. Only 1 sample is not enough to conclusively comment on the effect of lyophilization.

Response: We agree that the data from donor samples is insufficient to draw firm conclusions and have stated this as a weakness of our study (Discussion, line 717-721). We were constrained as the recruitment of donors was dependent on their willingness (in the form of a written consent) to participate in this study. 

8. Why PMPs are unlikely to be extracted by SMA? This information should be provided at the relevant place.

Response: We have mentioned the characteristics of PMPs in the Discussion (lines 603-605) which most probably prevented their incorporation into SMALPs, considering the mechanism of SMA action. 

9. Authors should show what possible TNF-a inducers are present in their SMALPs, in addition to LAM.

Response: Apart from LAM, the lipoproteins LpqH and PstS1 were also found in our SMALPs. As we have mentioned in the Discussion (lines 687-689), both the lipoproteins are involved in the induction of TNF-a through their binding to ‘toll-like’ receptors present on the surface of macrophages. 

 

Reviewer #2: The authors present data describing attempts to isolate membrane proteins from Mycobacterium tuberculosis using SMA polymer. In general the paper is technically sound, with a couple possible exceptions (noted below). I have some suggestions for editing a revised manuscript.

Comment: Line 47 should reference current WHO numbers given that they have been released. I’m not convinced that ‘membrane proteins’ needs to be abbreviated (MP). This hurts readability. In fact, generally through the manuscript there is an over-reliance on abbreviations. I found myself constantly trying to remember which abbreviations meant what, especially because many are close to each other and are not 'standard'. If there is no hard cap on word limit, there is no reason to abbreviate so much. For readability, the formatting of tables could be improved. The headings of the table should at least be bolded.

Response: Thanks for these helpful suggestions and we regret the inconvenience caused by the abbreviations. We have now given the current WHO numbers (Introduction, lines 55-58) and reduced the number of abbreviations. We have also reformatted the Tables. 

Comment: Regarding the data itself, the quality of the gel images is poor. They look like they were not de-stained long enough prior to imaging, or that the running buffer was contaminated resulting in extremely high background. Many of the lanes also appear over-loaded. If these samples are still available they should be re-run and properly destained to obtain a more appropriate signal to background.

Response: Thanks for pointing this out. As suggested, we have now re-run the samples with lesser sample loads and corresponding results are shown as Figure 3. Hopefully, they offer a better clarity. The diffused background staining in lower half of the SMA lanes persists even after extensive destaining. This is because SMA itself gets stained with coomassie blue (Reference 14). We have now clarified this in the legend to Figure 3.

Comment: The reason for lyophilising the samples is not really explained in the manuscript. What were the authors hoping to achieve by doing that? This is sort of touched on around line 510, but should be explained when the experiment is first introduced. Otherwise the reader is left a bit confused about why that choice was made. Do the authors have any evidence that these proteins will be functional after lyophilisation? What percent of proteins can be reproducibly assayed after lyophilisation – does that number actually change with SMALP-ing?

Response: Thanks for pointing this out. The reason for lyophilising the samples is now stated in revised Introduction (lines 130-134). Although the lyophilised MtM-SMALPs produced a T cell response which was comparable with their ‘wet’ counterparts, the data is insufficient to draw a firm conclusion. We have stated this as a weakness of our study (Discussion, lines 717-721). Nevertheless, since proteins within SMALPs have been shown to be functionally active (Introduction, 3rd paragraph) we have reasons to believe that their lyophilised counterparts will also be active, at least as antigens. The protein profiles before and after lyophilisation are a part of Figure 3A. 

Comment: It is not clear how many replicates are presented in figure 4? If this is only one replicate, it probably should not be included in the manuscript.

Response: As mentioned in the Materials and Methods (lines 260-261, 270-271), it is the average of two replicates. We have now also stated this in the legend to Figure 5 (earlier Fig. 4).

Comment: For the data around line 430. Was a SMALP-only control conducted? Do we know the response is not to the SMA polymer itself? Was the MtM sample treated the same way otherwise as the SMA such that the samples are matched for any other buffers/reagents?

Response: Yes, SMALP-only control was included (please see Material and Methods, lines 277-278; and Fig. 6A and 6B). Also, the samples were matched for buffers/reagents.

Comment: In the discussion, the authors posit that being a lipoprotein is why LpqH is more abundant in SMALPs. Were other lipoproteins similarly abundant? There are many lipoproteins in M. tuberculosis. If not, then the lipidation is not the reason for better extraction into SMALPs, and this should be re-written. Is LpqH abnormally abundant?

Response: As pointed out, there are close to 100 ‘putative’ lipoproteins in M. tuberculosis genome. However, so far only a few of them (LpqH, LprG, LprA and PstS1) have been proven experimentally; two of which (LpqH and PstS1, both being present in MtM-SMALPs) have been studied more extensively than the others (Ref 49 and Harding and Boom, Nature Reviews Microbiology, 2010, 8:296). The paucity of experimental data on lipoproteins of Mtb makes it difficult to say whether LpqH is abnormally abundant.

 

Response to Academic Editor

Comment: Both reviewers had concerns with figures, in particular TEM images and the clarity of the discoid shapes and how they are altered with SMALPs preparation methods. Also please address the gel image quality issues pointed out by reviewer 2. Reviewer 2 made note of the extensive use of abbreviations and the difficulty it creates for the reader. Please edit and make judicious use of abbreviations. Please address issues of clarity and rationale in the text identified by the reviewers. 

Response: Thanks for these helpful suggestions. We have addressed all of these concerns in our response to reviewers and have revised the manuscript accordingly.

---

## [Editor Report · Decision Letter 1]

20 Dec 2022

Immunochemical characterisation of styrene maleic acid lipid particles prepared from Mycobacterium tuberculosis plasma membrane

PONE-D-22-27916R1

Dear Dr. Sinha,

We’re pleased to inform you that your manuscript has been judged scientifically suitable for publication and will be formally accepted for publication once it meets all outstanding technical requirements.

Kind regards,

Christopher W Reid, Ph.D

Academic Editor

PLOS ONE

---

## [Editor Report · Acceptance letter]

28 Dec 2022

PONE-D-22-27916R1 

Immunochemical characterisation of styrene maleic acid lipid particles prepared from *Mycobacterium tuberculosis* plasma membrane 

Dear Dr. Sinha:

I'm pleased to inform you that your manuscript has been deemed suitable for publication in PLOS ONE. Congratulations! Your manuscript is now with our production department. 

Kind regards, 

on behalf of

Dr. Christopher W Reid 

Academic Editor

PLOS ONE